# Efficient Hamiltonian learning from equilibrium states

Adam Artymowicz

*California Institute of Technology, Pasadena, CA 91125, USA*

September 23, 2024

**Abstract**

We describe a novel algorithm that learns a Hamiltonian from local expectations of its Gibbs state using the free energy variational principle. The algorithm avoids the need to compute the free energy directly, instead using efficient estimates of the derivatives of the free energy with respect to perturbations of the state. These estimates are based on a new entropy bound for Lindblad evolutions, which is of independent interest. We benchmark the algorithm by performing black-box learning of a nearest-neighbour Hamiltonian on a 100-qubit spin chain. A implementation of the algorithm with a Python front-end is made available for use.

## 1 Introduction

In this work we consider the problem of learning a quantum Hamiltonian from its Gibbs state. Gibbs states are ubiquitous in quantum physics and represent systems in thermal equilibrium. They can be defined by a variational principle: they are the minimizers of free energy. This fact can in principle be used for Hamiltonian learning [1, 2], but a naive implementation of this idea is impractical because the free energy is known to be classically exponentially difficult to compute [3]. However, this does not preclude use of the variational principle in an efficient algorithm. Indeed, since the free energy is convex, the global variational principle is equivalent to a local one, which requires only knowledge of the derivatives of the free energy with respect to perturbations of the state. Linear response theory relates derivatives of free energy to locally measurable quantities, suggesting that they can be estimated without knowledge of the free energy itself. If such estimates can be made efficient, then the variational principle can be efficiently implemented.

The current work contains two main contributions. The first is a new lower bound on the entropy change due to a local perturbation of a quantum state (Theorem 1). We relate this to a hierarchy of semidefinite constraints known as the matrix EEB inequality [4] by showing that it is a relaxation of the free energy variational principle which can be enforced using polynomial classical resources.

Second, we use this to formulate a semidefinite algorithm for Hamiltonian learning. The algorithm either finds a local Hamiltonian respecting the matrix EEB inequality, or else it gives a proof that the given state is not a Gibbs state of any local Hamiltonian. We benchmark the algorithm by performing black-box learning of a nearest-neighbour Hamiltonian on 100 qubits from local expectation values with realistic levels of measurement noise.

These contributions are significant for several reasons. Hamiltonian learning is relevant in experimental settings [5]: near-term applications include studying frustrated systems by learning effective Hamiltonians [6] and probing entanglement properties of many-body states via their entanglement Hamiltonians [7]. For such applications, the problem of learning a Hamiltonian from local expectation values currently presents a practical bottleneck limiting the system sizes under consideration.

Indeed, local expectation values of systems of $\sim$100 qubits can in many cases be obtained either numerically or using current NISQ technology. Meanwhile, practical algorithms for Hamiltonian learning that do not assume prior information or additional control over the state have so far achieved reliable learning only for systems of 10 or fewer qubits. The algorithm we propose in this work has the potential to remove this bottleneck.

The entropy lower bound in Theorem 1 is also of independent interest beyond Hamiltonian learning. For instance, efficient preparation of Gibbs states on a quantum computer remains an important open problem. A common approach is to thermalize an initial state using Lindbladian dynamics. The convergence speed can be related to the rate of change of free energy, which can be estimated using Theorem 1.

We begin in Section 2 by introducing the main entropy lower bound (Theorem 1) and the matrix EEB inequality (Corollary 1). Section 3 describes the semidefinite algorithm.. In Section 4 we describe the results of numerical simulations on a 100-qubit spin chain. We conclude in Section 5 with a discussion of the results, and some future directions for research. This work additionally includes three appendices. In appendix A we compare our algorithm to some existing approaches, with a focus on practical performance. In appendix B we prove in detail the main theoretical results introduced in Section 2. Finally, C contains extra details about the numerical implementation described in section 4.

The implementation of the learning algorithm used in this work is available for use at

`https://github.com/artymowicz/hamiltonian-learning`

Aside from the learning algorithm itself, this repository also contains all routines used for performing the numerical tests in Section 4.

## 2 The lower bound on dS

In this section we begin by introducing Theorem 1 and derive the matrix EEB inequality as a consequence. We defer all proofs in this section to Appendix B. Let $\mathcal{H}$ be the Hilbert space of a quantum system, and assume that $\dim \mathcal{H} < \infty$. Let $\mathcal{A}$ be the set of all operators on $\mathcal{A}$. As a rule, we will use lowercase letters to denote elements of $\mathcal{A}$. We will denote the adjoint of an operator $a$ by $a^*$. We say a mixed state represented by a density matrix $\rho$ is *faithful* if the matrix $\rho$ is invertible. Let $\rho$ be such a state.

We are interested in perturbations of $\rho$ due to interactions with its environment. In the Lindblad formalism [8, 9] the evolution of the state $\rho$ under open-system dynamics is given for $t \geq 0$ by

$$\rho_t = e^{tL}[\rho] \tag{1}$$

where $L$ is the Lindbladian superoperator. Under the assumption that the environment is Markovian and interacts weakly with the system, $L$ can be written in the form

$$L[\rho] := \sum_{i,j=1}^{r} \left\{ \frac{1}{2} \boldsymbol{M}_{ij} [a_j^* a_i, \rho] + \boldsymbol{\Lambda}_{ij} (a_i \rho a_j^* - \frac{1}{2}(a_j^* a_i \rho + \rho a_j^* a_i)) \right\} \tag{2}$$

where $\boldsymbol{M}$ anti-Hermitian, $\boldsymbol{\Lambda}$ is positive-semidefinite, and $a_1, \ldots, a_r$ is a set of operators satisfying $\mathrm{tr}(\rho a_i^* a_j) = \delta_{ij}$ and $\mathrm{tr}(\rho a_i) = 0$ for all $i = 1, \ldots r$.

A straightforward calculation using the cyclic property of the trace yields the following expression for the first-order change in the expectation value of an observable under the evolution generated by the Lindbladian (2):

**Lemma 1.** *Let $h \in \mathcal{A}$ be selfadjoint and define the $r \times r$ matrix $\boldsymbol{H}_{ij} := \mathrm{tr}(\rho a_i^*[h, a_j])$. Then we have*

$$\frac{d}{dt}\bigg|_{t=0} \mathrm{tr}(\rho_t h) = \mathrm{tr}(\boldsymbol{M}\boldsymbol{H}_-) + \mathrm{tr}(\boldsymbol{\Lambda}\boldsymbol{H}_+) \tag{3}$$

*where $\boldsymbol{H}_\pm := (\boldsymbol{H} \pm \boldsymbol{H}^\dagger)/2$.*

Notice that if $a_1, \ldots, a_r$ and $h$ are operators of bounded locality, then the expression (3) uses only expectation values of operators of bounded locality. One may ask if a similar expression exists for the first-order change in the von Neumann entropy $S(\rho) = -\mathrm{tr}(\rho \log \rho)$. In general, this cannot be expected, since entropy is not a local property of the state. However, instead of an exact expression, the following proposition bounds first-order change in entropy using only the correlations of the hopping operators:

**Theorem 1.** *We have*

$$\frac{d}{dt}\bigg|_{t=0} S(\rho_t) \geq -\mathrm{tr}(\boldsymbol{\Lambda} \log \boldsymbol{\Delta}) \tag{4}$$

*where the matrix $\boldsymbol{\Delta}$ is defined as $\boldsymbol{\Delta}_{ij} := \mathrm{tr}(\rho a_j a_i^*)$.*

Let us remark on some ambiguities in the expression (2) and their effect on the above inequality. For a given Lindbladian $L$, the operators $a_1, \ldots, a_r$ and the matrices $\boldsymbol{M}$ and $\boldsymbol{\Lambda}$ in (2) are not uniquely defined. Indeed, there are two ambiguities in their definition[1]:

1. Applying a change of basis $a_i' = \sum_{ij} Q_{ij} a_j$ that preserves the condition $\mathrm{tr}(\rho a_i^* a_j) = \delta_{ij}$ and applying the inverse change of basis to the matrices $\boldsymbol{M}$ and $\boldsymbol{\Lambda}$.

2. Appending additional operators $a_{r+1}, \ldots, a_{r+q}$ to the list and setting all new matrix elements in $\boldsymbol{M}$ and $\boldsymbol{\Lambda}$ to zero.

While the first ambiguity does not change the right-hand side of (4), it turns out that the second ambiguity does. Indeed, adding operators to the list $a_1, \ldots, a_r$ increases the right-hand side of (4). This can be seen as the result of the nonlinearity of the matrix logarithm $\log(\boldsymbol{\Delta})$. We summarize the above discussion as follows. Let $\mathcal{P} = \mathrm{span}\{a_1, \ldots, a_r\} \subset \mathcal{A}$. Then the bound (4) depends only on $L$ and $\mathcal{P}$. Growing $\mathcal{P}$ improves the bound, but requires knowledge of a larger number of correlations.

In the remainder of this section we describe one of the consequences of the bound (4). The free energy of a state $\rho$ with respect to a Hamiltonian $h$ and a temperature $T$ is

$$F(\rho) := -TS(\rho) + \mathrm{tr}(\rho h). \tag{5}$$

Lemma 1 and Theorem 1 give an upper bound on the first-order change of free energy under the Lindbladian evolution generated by (2):

$$\frac{d}{dt}\bigg|_{t=0} F(\rho_t) \leq \mathrm{tr}(\boldsymbol{M}\boldsymbol{H}_-) + \mathrm{tr}(\boldsymbol{\Lambda}(T \log \boldsymbol{\Delta} + \boldsymbol{H}_+)) \tag{6}$$

Given a Hamiltonian $h$ and a temperature $T$, the Gibbs state $\rho := e^{-h/T}/\mathrm{tr}(e^{-h/T})$ is the unique minimizer of the free energy. Let us call a Lindbladian $L$ *$\mathcal{P}$-supported* if the operators $a_1, \ldots, a_r$ in the expression (2) can be chosen to lie in $\mathcal{P}$.

---

[1]It is shown in appendix B that these are the only ambiguities in the definition of $\boldsymbol{\Lambda}$.

**Corollary 1.** *If $\rho$ is the Gibbs state of a Hamiltonian $h$ then*

$$T \log \boldsymbol{\Delta} + \boldsymbol{H} \succeq 0. \tag{7}$$

*Moreover, if (7) fails, then there is a $\mathcal{P}$-supported Lindbladian that decreases the free energy of $\rho$ with respect to $h$.*

The inequality (7) is known as the matrix EEB inequality [4]. It is a hierarchy (depending on $\mathcal{P}$) of convex constraints that converges to the Hamiltonian of a Gibbs state.

## 3 Algorithm

In this section we apply the EEB inequality to the problem of learning the Hamiltonian of a Gibbs state. Given a set of selfadjoint traceless operators $h_1, \ldots, h_s$, we give an efficient algorithm that either finds a Hamiltonian in the span of $h_1, \ldots, h_s$ that satisfies the matrix EEB inequality, or else returns a Lindbladian that simultaneously rules out every $h$ in the span of $h_1, \ldots, h_s$ by decreasing the free energy.

By adding a regularization parameter to the matrix EEB inequality and setting $T = 1$ we get the following linear semidefinite program:

$$\underset{\substack{h \in \mathrm{span}\{h_1, \ldots, h_s\} \\ \mu \in \mathbb{R}}}{\text{minimize}} \qquad \mu \tag{8}$$

$$\text{subject to} \qquad \log(\boldsymbol{\Delta}) + \boldsymbol{H} + \mu I \succeq 0, \tag{9}$$

As before, $\boldsymbol{\Delta}$ and $\boldsymbol{H}$ are defined as

$$\boldsymbol{\Delta}_{ij} := \mathrm{tr}(\rho a_j a_i^*), \tag{10}$$

$$\boldsymbol{H}_{ij} := \mathrm{tr}(\rho a_i^* [h, a_j]). \tag{11}$$

where $a_1, \ldots, a_r$ satisfy $\mathrm{tr}(\rho a_i^* a_j) = \delta_{ij}$ and $\mathrm{tr}(\rho a_i) = 0$. The convex dual of the above program reads:

$$\underset{\substack{\boldsymbol{\Lambda} \succeq 0 \\ \boldsymbol{M}^\dagger = -\boldsymbol{M}}}{\text{maximize}} \qquad - \mathrm{tr}(\boldsymbol{\Lambda} \log(\boldsymbol{\Delta})) \tag{12}$$

$$\text{subject to} \qquad \mathrm{tr}(\boldsymbol{M} \boldsymbol{H}_-) + \mathrm{tr}(\boldsymbol{\Lambda} \boldsymbol{H}_+) = 0, \tag{13}$$

$$\mathrm{tr}(\boldsymbol{\Lambda}) = 1 \tag{14}$$

In light of Lemma 1 and Theorem 1, the dual program seeks a $\mathcal{P}$-supported Lindbladian that maximizes $dS/dt$ while preserving (to first order) the expectation values of the operators $h_1, \ldots, h_s$. Thus the dual program can be thought of as seeking the direction of steepest ascent for preparing the maximum-entropy state with prescribed expectation values of the operators $h_1, \ldots, h_s$.

A primal-dual pair is said to satisfy *Slater's condition* if the primal program has a strictly feasible point. Such a point can be found for (8) by setting $h = 0$ and taking $\mu$ sufficiently large. As a consequence, a standard result in convex optimization states that the optimal values of the primal and dual program coincide [10]. Thus we have:

**Proposition 1.** *Let $\mu$, $h$ be the optmizers of the primal program (8), and $\boldsymbol{M}, \boldsymbol{\Lambda}$ the optimizers of the dual program (12).*

1. *If $\mu \leq 0$, then $h$ satisfies the matrix EEB inequality (7).*

2. If $\mu > 0$ then the Lindbladian $L$ corresponding to $\mathbf{M}$ and $\mathbf{\Lambda}$ increases the entropy of $\rho$ while preserving $\mathrm{tr}(\rho h_\alpha)$ to first order for every $\alpha = 1, \ldots, s$.

Thus, by running the primal-dual pair, we are guaranteed to get either a Hamiltonian $h$ that satisfies the EEB inequality, or else get a $\mathcal{P}$-supported Lindbladian that a) acts as an interpretable guarantee that $\rho$ is not the Gibbs state of any Hamiltonian in the search space, and b) is a heuristic for the best available Lindbladian for thermalizing the state $\rho$.

We conclude this section by discussing the computational complexity of the primal-dual pair. Interior-point methods produce solutions to the primal and dual problem whose objectives are within $\epsilon$ of the true optimum in $poly(r) \log(1/\epsilon)$ time, where $r$ is the dimension of $\mathcal{P}$ [11]. In the remainder of the paper we will restrict our attention to a system of $n$ spins on a lattice, where it is natural to choose an integer $k > 0$ and let $\mathcal{P}$ be the span of all $k$-local Pauli operators, and the variational Hamiltonian terms $h_1, \ldots, h_s$ to be the set of all $k'$-local Pauli operators. Here we mean $k$-local in the sense that the operator acts trivially outside a set of $k$ *contiguous* qubits. With these choices we have $r = O(4^k n)$ and the algorithm requires $O(4^{k'+2k} n^2)$ expectation values of Paulis of weight at most $k + 2k$.

# 4    Numerical results

In this section, we describe a numerical implementation of the algorithm from section 3, applied to the problem of learning a nearest-neighbour 100-qubit Hamiltonian from local expectation values of its Gibbs state. A variable amount of noise was added to the input of the algorithm. At zero noise, this acts as a test to how tightly the matrix EEB inequality constrains the set of possible Hamiltonians. At nonzero noise levels, this acts as a test of the number of independent samples of the state $\rho$ required to accurately reconstruct the Hamiltonian.

**Learning an XXZ Hamiltonian**

The MPS purification technique [12] was used to prepare thermal states of the following anisotropic Heisenberg ferromagnet:

$$h_{XXZ} = -\sum_{i=1}^{n-1} (\sigma_i^x \sigma_{i+1}^x + \sigma_i^y \sigma_{i+1}^y + \frac{1}{2}\sigma_i^y \sigma_{i+1}^y). \tag{15}$$

with $n = 100$. Both the set of perturbing operators $b_1, \ldots, b_r$ and the set of variational Hamiltonian terms $h_1, \ldots, h_s$ were chosen to be the 1192 geometrically 2-local Pauli operators. Measurement error was simulated by adding Gaussian noise with variance $\sigma_{noise}$ to the expectation value of each Pauli operator. The learning algorithm itself was implemented in Python, using the MOSEK solver [13] for the semidefinite optimization[2].

Hamiltonian recovery error was quantified using the overlap as in [16]: let $y \in \mathbb{R}^s$ be the vector of recovered Hamiltonian coefficients and $z \in \mathbb{R}^s$ the vector of true Hamiltonian coefficients. The Hamiltonian recovery error is then defined as the relative angle of the two, which for small angles approximately equals the reciprocal of the signal-to-noise ratio:

$$\theta = \arccos\left(\frac{|\langle y|z\rangle|}{\|y\|\|z\|}\right) \approx \frac{\|y - z\|}{\|z\|}. \tag{16}$$

Note that this metric is not sensitive to the overall scaling of the Hamiltonian. This degree of freedom is effectively the inverse temperature $1/T$. Interestingly, the algorithm reconstructed the

---

[2]The convex modeling language CVXPy [14] and the open source solver SCS [15] were used in prototyping but not in the final code.

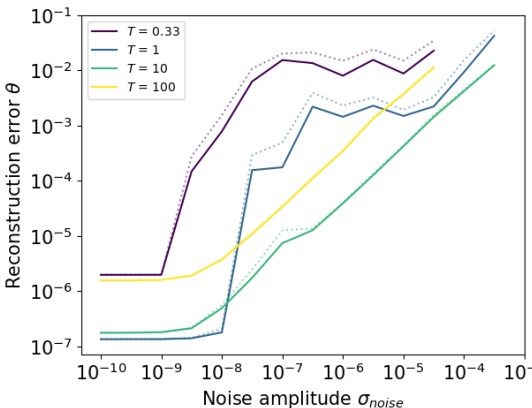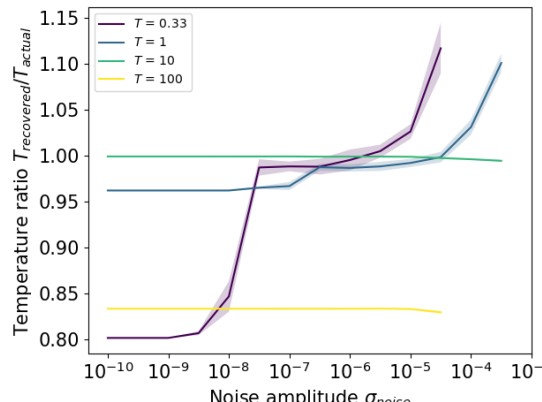

*Figure 1: Numerical results for the 100-qubit anisotropic Heisenberg model (15) at several temperatures.* Left: *Recovery error $\theta$ as a function of noise amplitude $\sigma_{noise}$, averaged over 10 runs. Dotted line is (mean) + (standard deviation).* Right: *Ratio of recovered temperature to actual temperature, averaged over 10 runs. Shaded region is (mean) $\pm$ (standard deviation).*

"projective" degrees of freedom of the Hamiltonian terms much more accurately than it did its overall scale (or equivalently, the temperature).

The Hamiltonian recovery error $\theta$ and the recovered temperature $T$ are plotted against $\sigma_{noise}$ in Figure 1. A temperature-dependent noise threshold is found between $\sigma_{noise} \approx 10^{-5}$ and $\sigma_{noise} \approx 10^{-3}$ above which the matrix $\boldsymbol{\Delta}$ ceases to be positive definite – these are the right endpoints of the plots in Figure 1. The algorithm could possibly be emended to work for higher noise values by projecting onto the positive eigenspace of $\boldsymbol{\Delta}$, but we leave this to future work.

As one shrinks the noise amplitude, the recovery error first decreases (for high temperatures, this decrease is linear to a good approximation). This persists up until, at some temperature-dependent critical value of the noise amplitude, the recovery error plateaus. We interpret this two-stage behaviour as follows. In the limit of zero measurement error, perfect recovery is not guaranteed because the matrix EEB inequality (7) is weaker than the Gibbs condition. Instead, it defines a convex set of candidate Hamiltonians, and the algorithm picks one of these by maximizing the regularization parameter $\mu$. The recovery error is then on the order of the diameter of this convex set. Thus for low enough levels of measurement noise the recovery error is approximately noise-independent.

The only way to lower the levels of these plateaux is to enlarge $\mathcal{P}$, which tightens the matrix EEB constraint. This is relevant if one wants to prove asymptotic bounds on the number of copies of the state and the computational resources needed to specify the Hamiltonian up to an arbitrarily low error. Such results, however, do not necessarily have practical implications. Indeed, for the particular Hamiltonian under consideration, the plateaux start at noise amplitudes $\sigma_{noise}$ of around $10^{-9}$ to $10^{-8}$. Assuming that expectation values are estimated from independent copies of the state, this would require on the order of $10^{16}$ to $10^{18}$ samples, far beyond what is experimentally feasible anyway. So for practical applications it may be more important to understand the high-noise regime rather than the locations of the plateaux.

# 5 Outlook

Let us conclude by describing some directions for future research. While Proposition 4 establishes the correctness of the algorithm, it suffers from two important limitations which must be overcome

if one is to prove sample complexity and computational complexity bounds. First, neither the feasibility nor the recoverability statements of Proposition 4 take into account measurement noise in the expectation values of the state, which is unavoidable whenever these are estimated using finitely many copies of the state. Second, the recoverability statement only holds when the set of perturbing operators is grown to a complete set of operators. The utility of this algorithm depends on approximate recoverability when the set of perturbing operators is far smaller than a complete set. Section 4 gives numerical evidence that this is indeed the case, but a proof is still lacking.

### Acknowledgements

The author would like to thank Anton Kapustin, Hsin-Yuan (Robert) Huang, and Elliott Gesteau for helpful discussions and comments on the draft.

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

# A    Comparison to other work

In this appendix we compare the Hamiltonian learning algorithm discussed in sections 3 and 4 to a few existing approaches, focusing on practical performance. We include only algorithms that learn from independent copies of an identical state $\rho$ without assuming any other control over $\rho$. On the theoretical side, the recent series of works [2, 17, 18] culminated with a proof by Bakshi et al. that this problem can be solved using polynomial classical resources [18]. However, their result is asymptotic and no implementation yet exists by which to assess its practical performance. Algorithms which have so far seen practical implementation either use exponential classical resources [2, 19, 20], or solve the more general problem of learning a Hamiltonian from a stationary state, which we show below is ill-posed in the setting of Gibbs states. Thus, so far a demonstration that the problem of learning a Hamiltonian from local expectations can reliably be solved in the noisy 100-qubit regime has been missing.

**Local equilibrium criteria**

On a theoretical level, the current work bears closest resemblance to the algorithm given by Bakshi et al in [18]. The *KMS condition* [21, Section 5.3] states that $\rho$ is the Gibbs state of a Hamiltonian $h$ at temperature $T = 1/\beta$ if and only if

$$\operatorname{tr}(\rho e^{-\beta H} a e^{\beta H} b) = \operatorname{tr}(\rho b a) \tag{17}$$

for all $a, b \in \mathcal{A}$. Another is the *EEB condition* [21, Theorem 5.3.15]: $\rho$ is the Gibbs state of $h$ at temperature $T = 1/\beta$ if and only if

$$\operatorname{tr}(\rho a^* a) \log \left( \frac{\operatorname{tr}(\rho a a^*)}{\operatorname{tr}(\rho a^* a)} \right) + \beta \operatorname{tr}(a^*[h, a]) \geq 0 \tag{18}$$

for all $a \in \mathcal{A}$. Both of these are local conditions in the sense that they can be checked for a subset of operators, yielding relaxations of the Gibbs condition. While the matrix EEB inequality used in the current work is a semidefinite relaxation of the EEB condition[4], the algorithm in [18] uses a sum-of-squares relaxation of the KMS condition.

**Learning from steady states**

The algorithms [22] and [23] learn a Hamiltonian from local expectation values of a steady state. This can be applied to a Gibbs state, since a Gibbs state is necessarily a steady state. Both these algorithms work by approximately enforcing the linear constraints

$$\operatorname{tr}(\rho[h, b_i]) = 0, \quad i = 1, \ldots, m \tag{19}$$

for some choice of operators $b_1, \ldots, b_m$.

A constraint of this form can be seen to be implicit in the constraint used in the current algorithm. Indeed, by breaking the regularized EEB constraint

$$\log(\boldsymbol{\Delta}) + \boldsymbol{H} + \mu \boldsymbol{1} \succeq 0 \tag{20}$$

into its hermitean and anti-hermitean parts, (20) can be see to be equivalent to the pair of constraints

$$\boldsymbol{H}_- = 0 \tag{21}$$
$$\log(\boldsymbol{\Delta}) + \boldsymbol{H}_+ + \mu \boldsymbol{1} \succeq 0, \tag{22}$$

where $\boldsymbol{H}_\pm = (\boldsymbol{H} \pm \boldsymbol{H}^\dagger)/2$. It is not hard to check that $\boldsymbol{H}_- = 0$ if and only if

$$\text{tr}(\rho[h, a_i^* a_j]) = 0 \text{ for all } 1 \leq i, j \leq r. \tag{23}$$

Let us remark on why the additional positive-semidefinite constraint is necessary. The linear constraint alone cannot recover $h$ if the state $\rho$ has any local symmetries other than the Hamiltonian. This happens, for instance, for Gibbs states of the XXZ model (15) considered in Section 4. A Gibbs state of $h_{XXZ}$ at any temperature will commute with the generator $q := \sum_i \sigma_i^z$ of onsite $z$-rotations. This means that the linear constraint alone, and in general any algorithm that does not discriminate steady states from Gibbs states, cannot distinguish the Hamiltonians $h_{XXZ} + \lambda q$ for different values of the parameter $\lambda$.

### Algorithms using loss functions

Several algorithms [2, 19, 20] work by minimizing a loss function which is intended to measure the discrepancy between the input state and the Gibbs state of a trial Hamiltonian. In [2] the loss function can be shown to equivalent to the relative entropy and requires computing the partition function of the trial Hamiltonian. In both [19] and [20], the loss function is a $\chi^2$ statistic based on a random measurement scheme. In all three cases, the loss function requires exponential classical resources to compute, preventing these algorithms from being scaled beyond the $\sim 10$ qubit regime.

## B   Proofs

In this appendix we prove the claims made in section 2. First we prove a standard form for Markovian Lindbladians analogous to the GKSL standard form [24, 8]. In what follows $\rho$ will always refer to a faithful state.

**Proposition 2** (Standard form for Lindbladians). *Every Markovian Lindbladian can be written in the form*

$$L[\sigma] = \sum_{i,j=1}^r \left\{ \frac{1}{2} \boldsymbol{M}_{ij} [a_j^* a_i, \sigma] + \boldsymbol{\Lambda}_{ij} (a_i \sigma a_j^* - \frac{1}{2}(a_j^* a_i \sigma + \sigma a_j^* a_i)) \right\} \tag{24}$$

*where $\boldsymbol{M}$ is anti-Hermitian, $\boldsymbol{\Lambda}$ is positive-semidefinite, and $a_1, \ldots, a_r$ satisfy $\text{tr}(\rho a_i^* a_j) = \delta_{ij}$ and $\text{tr}(\rho a_i) = 0$. For a given $L$, the matrix $\boldsymbol{\Lambda}$ is uniquely determined by the choice of $a_1, \ldots, a_r$.*

Next we prove a detailed version of the entropy bound in Theorem 1:

**Proposition 3** (Entropy bound). *Let $L$ be a Lindbladian of the form given by Proposition 2 and define*

$$\mathscr{S} := -\text{tr}(\boldsymbol{\Lambda} \log \boldsymbol{\Delta}), \tag{25}$$

*where $\boldsymbol{\Delta}_{ij} := \text{tr}(\rho a_j a_i^*)$. Then*

*i)* $\mathcal{S}$ *depends only on* $\rho$, $L$, *and* $\mathcal{P} = \text{span}\{a_1, \ldots, a_r\}$.

*ii)* *If* $\mathcal{P} \subset \mathcal{P}'$ *then* $\mathcal{S}(\rho, L, \mathcal{P}) \leq \mathcal{S}(\rho, L, \mathcal{P}')$.

*iii)* *If* $\mathcal{P} = \{a \in \mathcal{A} : \text{tr}(\rho a) = 0\}$ *then*

$$\mathscr{S} = \frac{d}{dt}\bigg|_{t=0} S(\rho_t) \tag{26}$$

*for* $\rho_t = e^{tL}[\rho]$.

Finally, we state in detail the matrix EEB inequality.

**Proposition 4** (Matrix EEB inequality). *Let* $a_1, \ldots, a_r$ *be operators satisfying* $\text{tr}(\rho a_i^* a_j) = \delta_{ij}$ *and* $\text{tr}(\rho a_i)$. *For a given* $T \geq 0$ *define* $K$ *to be the convex set of all traceless selfadjoint operators* $h \in \mathcal{A}$ *such that*

$$T \log(\boldsymbol{\Delta}) + \boldsymbol{H} \succeq 0 \tag{27}$$

*where* $\boldsymbol{\Delta}_{ij} := \text{tr}(\rho a_j a_i^*)$ *and* $\boldsymbol{H} = \text{tr}(\rho a_i^*[h, a_j])$. *Then*

*i)* $K$ *depends only on* $\rho$, $T$, *and* $\mathcal{P} := \text{span}\{a_1, \ldots, a_r\}$.

*ii)* *If* $\mathcal{P} \subset \mathcal{P}'$ *then* $K(\rho, T, \mathcal{P}') \subset K(\rho, T, \mathcal{P})$.

*iii)* *If* $\mathcal{P} = \{a \in \mathcal{A} : \text{tr}(\rho a) = 0\}$ *then* $K(\rho, T, \mathcal{P})$ *is a singleton containing the unique traceless operator* $h$ *such that* $\rho = e^{-h/T} / \text{tr}(e^{-h/T})$.

Parts *iii)* of the above two propositions can be seen as convergence results. We remark however that when $\mathcal{P} = \{a \in \mathcal{A} : \text{tr}(\rho a) = 0\}$, the expressions (25) and (27) use the expectation values of all operators, and thus requires full tomography of the state $\rho$. As such, this result does not give a practical convergence proof for the Hamiltonian learning algorithm considered in section 3. Instead it acts as a sanity check that the algorithm performs no worse than the naive algorithm using full state tomography.

The proofs of these three propositions will use the Gelfand-Naimark-Segal (GNS) construction, which we introduce briefly now. Although our introduction is entirely self-contained, readers wanting more details are referred to the standard references [25, 26], or to the lecture notes [27] which contain an introduction aimed at physicists.

Let $\mathcal{A}$ be the space of all operators on the physical Hilbert space $\mathcal{H}$, and let $\rho$ be a faithful state. The bilinear form $(a, b) \mapsto \text{tr}(\rho a^* b)$ endows $\mathcal{A}$ with the structure of a Hilbert space. For an operator $a \in \mathcal{A}$ we write $|a\rangle$ when we view $a$ as a vector in this Hilbert space[3]. The GNS vector $|1\rangle$ corresponding to the identity operator is usually denoted $|\Omega\rangle$.

The *modular operator* $\Delta : \mathcal{A} \to \mathcal{A}$ is defined by the equation

$$\langle a|\Delta|b\rangle = \text{tr}(\rho b a^*). \tag{28}$$

It is easy to check that $\Delta$ is self-adjoint with respect to the GNS inner product. As the following calculation shows, an equivalent characterization of $\Delta$ is that it takes a GNS vector $|b\rangle$ to $|\rho b \rho^{-1}\rangle$:

$$\langle a|\Delta|b\rangle = \text{tr}(\rho b a^*) \tag{29}$$

$$= \text{tr}((\rho b \rho^{-1})\rho a^*) \tag{30}$$

$$= \text{tr}(\rho a^*(\rho b \rho^{-1})) \tag{31}$$

$$= \langle a|\rho b \rho^{-1}\rangle. \tag{32}$$

---

[3] There is a close analogy between this notation and the state-operator correspondence in CFT.

We will use the following conventions: operators on the physical Hilbert space will be denoted by lowercase letters, operators on the GNS Hilbert space will be denoted by capital letters like $\Delta$ and $\Lambda$, and numerical matrices will be denoted by boldface captial letters like $\boldsymbol{\Delta}$ and $\boldsymbol{\Lambda}$.

**Proof of Proposition 2**

First let us show that every Markovian Lindbladian has a parametrization of the form (24). The GKSL theorem [24, 8] says that every Markovian Lindbladian has an expression of the form (24) but where $a_1, \ldots, a_r$ don't necessarily satisfy $\mathrm{tr}(\rho a_i^* a_j) = \delta_{ij}$ and $\mathrm{tr}(\rho a_i) = 0$. Notice that the right-hand side of (24) is invariant under the following two operations

1. Applying a coordinate transformation $a_i \mapsto \sum_j \boldsymbol{Q}_{ij} a_j$ while taking $\boldsymbol{\Lambda} \mapsto (\boldsymbol{Q}^{-1})^\dagger \boldsymbol{\Lambda} \boldsymbol{Q}^{-1}$ and $\boldsymbol{M} \mapsto (\boldsymbol{Q}^{-1})^\dagger \boldsymbol{M} \boldsymbol{Q}^{-1}$.

2. Adding new operators $a_{r+1}, \ldots, a_{r+q}$ to the list and setting all new matrix elements of $\boldsymbol{M}$ and $\boldsymbol{\Lambda}$ to zero.

Using the above operations we can ensure that $r = N^2$ (where $N$ is the dimension of the physical Hilbert space $\mathcal{H}$), $a_{N^2} = 1_{\mathcal{H}}$, and $\mathrm{tr}(\rho a_i^* a_j) = \delta_{ij}$ for $1 \leq i, j \leq N^2$. Notice now that for every term in (24) where $i = N^2$, the dissipative part can be absorbed into the unitary part:

$$\frac{1}{2} \boldsymbol{M}_{N^2, j}[a_j^*, \sigma] + \boldsymbol{\Lambda}_{N^2, j}(\sigma a_j^* - \frac{1}{2}(a_j^* \sigma + \sigma a_j^*)) = \frac{1}{2}(\boldsymbol{M}_{N^2, j} - \boldsymbol{\Lambda}_{N^2, j})[a_j^*, \sigma]. \tag{33}$$

Together with an analogous calculation for terms where $j = N^2$, we have

$$L[\sigma] = [h, \sigma] + \sum_{i,j=1}^{N^2-1} \boldsymbol{\Lambda}_{ij}(a_i \sigma a_j^* - \frac{1}{2}(a_j^* a_i \sigma + \sigma a_j^* a_i)) \tag{34}$$

where

$$h = \frac{1}{2} \sum_{i,j=1}^{N^2} (\boldsymbol{M}_{ij} + (\delta_{i,N^2} - \delta_{j,N^2})\boldsymbol{\Lambda}_{ij}). \tag{35}$$

To conclude the existence proof, we need to replace $h$ with $\sum_{i,j=1}^{N^2-1} \boldsymbol{M}'_{ij} a_i^* a_j$ for some antiselfadjoint matrix $\boldsymbol{M}'_{ij}$. We will use the following lemma:

**Lemma 2.** *Every $a \in \mathcal{A}$ can be written as $a = \sum_{i=1}^m b_i^* c_i + \gamma 1_{\mathcal{H}}$ where $\mathrm{tr}(\rho b_i) = \mathrm{tr}(\rho c_i) = 0$ and $\gamma \in \mathbb{C}$.*

*Proof.* For $N = 1$ this is immediate with $m = 0$ and $\gamma = a$. Suppose that $N > 1$. Let $\rho = \sum_{i=1}^N \rho_i |i\rangle\langle i|$. It suffices to prove the claim for $a = |i\rangle\langle j|$ for any $1 \leq i, j \leq N$. Choose any $k \neq j$. Then we have $a = b^* c$ where $b = |j\rangle\langle i|$ and $c = |j\rangle\langle j| - \frac{\rho_j}{\rho_k}|k\rangle\langle k|$. $\square$

Since $a_1, \ldots, a_{N^2-1}$ form a basis for $\{a \in \mathcal{A} : \mathrm{tr}(\rho a) = 0\}$, by the above Lemma we can write

$$h = \sum_{i,j=1}^{N^2-1} c_{ij} a_i^* a_j + \gamma 1_{\mathcal{H}} \tag{36}$$

$$= \frac{1}{2} \sum_{i,j=1}^{N^2-1} (c_{ij} - \overline{c_{ji}}) a_i^* a_j + Im(\gamma) 1_{\mathcal{H}} \tag{37}$$

for some $\{c_{ij}\}_{i,j=1}^{N^2-1}$, where the second line is because $h$ is anti-Hermitian. Thus finally we have

$$L[\sigma] = \frac{1}{2} \sum_{i,j=1}^{N^2-1} (c_{ij} - \overline{c_{ji}})[a_i^* a_j, \sigma] + \sum_{i,j=1}^{N^2-1} \mathbf{\Lambda}_{ij}(a_i \sigma a_j^* - \frac{1}{2}(a_j^* a_i \sigma + \sigma a_j^* a_i)) \tag{38}$$

which establishes the existence statement of Proposition 2.

The uniqueness statement will follow from the following result, which we will also use in the proof of Theorem 1. Let $\Lambda : \mathcal{A} \to \mathcal{A}$ be the positive-semidefinite operator

$$\Lambda := \sum_{i,j=1}^{r} \mathbf{\Lambda}_{ij} |a_i\rangle\langle a_j|. \tag{39}$$

**Lemma 3.** *Suppose a Lindbladian $L$ is expressed in the form (24) where the operators $a_1, \ldots, a_r$ are unrestricted. Let $Q = 1 - |\Omega\rangle\langle\Omega|$.*

  *i) The operator $Q\Lambda Q$ is independent of parametrization, ie. for a given $L$ it does not depend on $(a_1, \ldots, a_r, \mathbf{M}, \mathbf{\Lambda})$.*

  *ii) If $a_1, \ldots, a_r$ are required to satisfy $\mathrm{tr}(\rho a_i) = 0$, then $\Lambda = Q\Lambda Q$ and thus $\Lambda$ is parametrization-independent.*

*Proof.* To prove part *i*), suppose $(a_1, \ldots, a_r, \mathbf{M}, \mathbf{\Lambda})$ and $(a_1', \ldots, a_{r'}', \mathbf{M}', \mathbf{\Lambda}')$ are two parametrizations of $L$ of the form (24). Notice that $\Lambda$ is invariant under the operations 1. and 2. above. As a result we can assume without loss of generality that $r = r' = N^2$ and $a_1, \ldots, a_{N^2} = a_1', \ldots, a_{N^2}'$ is a basis of $\mathcal{A}$ with $a_{N^2} = 1_{\mathcal{H}}/2^{N-1}$ and $\mathrm{tr}(a_i^* a_j) = \delta_{ij}$ for $1 \leq 1, j \leq N^2$. Then the trick used to obtain (34) shows that there are self-adjoint operators $h$ and $h'$ such that

$$L[\sigma] = -i[h, \sigma] + \sum_{i,j=1}^{N^2-1} \mathbf{\Lambda}_{ij}(a_i \sigma a_j^* - \frac{1}{2}(a_j^* a_i \sigma + \sigma a_j^* a_i)) \tag{40}$$

$$= -i[h', \sigma] + \sum_{i,j=1}^{N^2-1} \mathbf{\Lambda}_{ij}'(a_i \sigma a_j^* - \frac{1}{2}(a_j^* a_i \sigma + \sigma a_j^* a_i)). \tag{41}$$

for every $\sigma$. By adding a multiple of the identity, $h$ and $h'$ can be made traceless and so the uniqueness statement of Theorem 2.2 in [24] shows that $\mathbf{\Lambda}_{ij} = \mathbf{\Lambda}_{ij}'$ for all $1 \leq i, j \leq N^2 - 1$. Thus we have

$$\Lambda' - \Lambda = \sum_{i=N^2 \text{ or } j=N^2} (\mathbf{\Lambda}_{ij}' - \mathbf{\Lambda}_{ij}) |a_i\rangle\langle a_j| \tag{42}$$

$$= |\Omega\rangle\langle a| + |a\rangle\langle\Omega|. \tag{43}$$

for some $a \in \mathcal{A}$. The result then follows from the fact that $Q(|\Omega\rangle\langle a| + |a\rangle\langle\Omega|)Q = 0$.

Part *ii*) then follows immediately from part *i*) and the fact that $\mathrm{tr}(\rho a_i) = \langle\Omega|a_i\rangle$. $\qquad\square$

The above lemma proves the uniqueness statement of Proposition 2, since the additional condition $\mathrm{tr}(\rho a_i^* a_j) = \delta_{ij}$ implies that $\mathbf{\Lambda}_{ij} = \langle a_i|\Lambda|a_j\rangle$.

**Proof of Proposition 3**

Part $i$).

Let $Q : \mathcal{A} \to \mathcal{A}$ be the orthogonal projection onto $\mathcal{P} := \operatorname{span}\{a_1, \ldots, a_r\}$. Since $\boldsymbol{\Delta}$ is the coordinate expression for $Q\Delta Q$ in the basis $a_1, \ldots, a_r$ and since $\langle a_i | a_j \rangle = \operatorname{tr}(\rho a_i^* a_j) = \delta_{ij}$ it is easy to check that

$$- \operatorname{tr}(\boldsymbol{\Lambda} \log(\boldsymbol{\Delta})) = - \operatorname{tr}(\Lambda Q \log(Q\Delta Q) Q) \tag{44}$$

By part $ii$) of Lemma 3, this expression depends only on $L, \rho$, and $\mathcal{P}$.

Part $ii$).

We will use the operator version of Jensen's inequality applied to the matrix logarithm, which follows from the main theorem in [28] and the fact that log is operator convex [29]:

**Lemma 4** (Operator Jensen's inequality for log). *Let $\mathcal{K}$ be a Hilbert space and let $Q$ and $Q'$ be projections in $\mathcal{K}$ such that the image of $Q$ is contained in the image of $Q'$. Then for any positive operator $M$ on $\mathcal{K}$ we have*

$$Q \log(QMQ) Q \succeq Q \log(Q'MQ') Q. \tag{45}$$

Let $(a_1, \ldots, a_r, \boldsymbol{M}, \boldsymbol{\Lambda})$ and $(a_1', \ldots, a_r', \boldsymbol{M}', \boldsymbol{\Lambda}')$ be two parametrizations of $L$, and suppose $\mathcal{P} \subset \mathcal{P}'$. Letting $Q$ and $Q'$ be the orthogonal projections onto $\mathcal{P}$ and $\mathcal{P}'$, we have

$$- \operatorname{tr}(\boldsymbol{\Lambda} \log(\boldsymbol{\Delta})) = - \operatorname{tr}(\Lambda Q \log(Q\Delta Q) Q) \tag{46}$$
$$\leq - \operatorname{tr}(\Lambda Q \log(Q'\Delta Q') Q) \tag{47}$$
$$= - \operatorname{tr}(\Lambda Q' \log(Q'\Delta Q') Q') \tag{48}$$
$$= - \operatorname{tr}(\boldsymbol{\Lambda}' \log(\boldsymbol{\Delta}')) \tag{49}$$

where in the third line we used the fact that $\Lambda = Q\Lambda Q = Q'\Lambda Q'$.

Part $iii$).

We begin by computing a general expression for the derivative of the entropy:

**Lemma 5.** *Let $\sigma_t$, $t \geq 0$ be a smooth path of density matrices and suppose that $\sigma_0$ is faithful. Write $S_t = - \operatorname{tr}(\sigma_t \log \sigma_t)$. Then using a prime to denote a time-derivative at $t = 0$, we have*

$$S' = - \operatorname{tr}(\sigma' \log(\sigma)). \tag{50}$$

*Proof.* We have

$$- \operatorname{tr}(\sigma \log(\sigma))' = - \operatorname{tr}(\sigma' \log(\sigma)) - \operatorname{tr}(\sigma \log(\sigma)'). \tag{51}$$

Using the power series of log about the identity operator and the cyclicity of the trace, the second term can be seen to equal $- \operatorname{tr}(\rho') = 0$. $\qquad \square$

Now we apply the above lemma to the time-evolution of our state $\rho$ generated by the Lindbladian (24):

**Lemma 6.** *With $\rho_t = e^{tL}[\rho]$, we have*

$$\left. \frac{d}{dt} \right|_{t=0} S(\rho_t) = - \operatorname{tr}(\Lambda \log(\Delta)) \tag{52}$$

*where $\Lambda := \sum_{ij} \boldsymbol{\Lambda}_{ij} |a_j\rangle\langle a_i|$*

*Proof.* Since the first term of (24) generates a unitary evolution it does not contribute to the entropy and so we may set $\boldsymbol{M} = 0$ without loss of generality. Since $\Delta|a\rangle = |\rho a \rho^{-1}\rangle$, we have $\log(\Delta)|a\rangle = |[\log(\rho), a]\rangle$. Thus we can expand the right-hand of (52) as

$$-\operatorname{tr}(\Lambda \log(\Delta)) = -\sum_{ij} \boldsymbol{\Lambda}_{ij} \langle a_j| \log(\Delta)|a_i\rangle \tag{53}$$

$$= -\sum_{ij} \boldsymbol{\Lambda}_{ij} \langle a_j|[\log(\rho), a_i]\rangle \tag{54}$$

$$= -\sum_{ij} \boldsymbol{\Lambda}_{ij} \operatorname{tr}(\rho a_j^*[\log(\rho), a_i]) \tag{55}$$

$$= -\sum_{ij} \boldsymbol{\Lambda}_{ij} \left[ \operatorname{tr}(a_i \rho a_j^* \log(\rho)) - \operatorname{tr}(\rho \log(\rho) a_j^* a_i) \right] \tag{56}$$

$$= -\sum_{ij} \boldsymbol{\Lambda}_{ij} \left[ \operatorname{tr}(a_i \rho a_j^* \log(\rho)) - \frac{1}{2}\operatorname{tr}(a_j^* a_i \rho \log(\rho)) - \frac{1}{2}\operatorname{tr}(\rho a_j^* a_i \log(\rho)) \right] \tag{57}$$

$$= -\operatorname{tr}(L[\rho] \log(\rho)), \tag{58}$$

which equals $\frac{d}{dt}\big|_{t=0} S(\rho_t)$ by Lemma 5. $\qquad\square$

Finally, we need the following lemma:

**Lemma 7.** *Let $Q = 1 - |\Omega\rangle\langle\Omega|$ be the orthogonal projection onto $\{a \in \mathcal{A} : \operatorname{tr}(\rho a) = 0\}$. We have*

$$Q \log(Q\Delta Q)Q = \log(\Delta). \tag{59}$$

*Proof.* Since $\Delta|\Omega\rangle = |\Omega\rangle$, we have $\log(\Delta)|\Omega\rangle = 0$, which proves the lemma. $\qquad\square$

We are now ready to prove part iii). Suppose $\mathcal{P} = \{a \in \mathcal{A} : \operatorname{tr}(\rho a) = 0\}$. Then

$$-\operatorname{tr}(\boldsymbol{\Lambda} \log(\boldsymbol{\Delta})) = -\operatorname{tr}(\Lambda Q \log(Q\Delta Q)Q) \tag{60}$$

$$= -\operatorname{tr}(\Lambda \log(\Delta)) \tag{61}$$

$$= \frac{d}{dt}\bigg|_{t=0} S(\rho_t). \tag{62}$$

**Proof of Proposition 4**

Parts $i$) and $ii$) follow from the corresponding parts of Proposition 3.

Part $iii$).

Suppose first that $\rho = e^{-h/T}/\operatorname{tr}(e^{-h/T})$. Define $H : \mathcal{A} \to \mathcal{A}$ as $H : |a\rangle \to |[h, a]\rangle$. Since $T \log(\Delta) + H = 0$, Lemma 7 gives $TQ \log(Q\Delta Q)Q + H = 0$, and sandwiching this expression between $\langle a_i|$ and $|a_j\rangle$ for all $1 \le i, j \le N^2 - 1$ gives $T \log(\boldsymbol{\Delta}) + \boldsymbol{H} = 0$.

Conversely, suppose $h$ satisfies the matrix EEB inequality for $\mathcal{P} = \{a \in \mathcal{A} : \operatorname{tr}(\rho a) = 0\}$. This implies in particular that $\boldsymbol{H}$ is self-adjoint, and the calculation

$$0 = \boldsymbol{H}_{ij} - \overline{\boldsymbol{H}}_{ji} \tag{63}$$

$$= \operatorname{tr}([\rho, h] a_i^* a_j) \tag{64}$$

together with Lemma 2 shows that $[h, \rho] = 0$. From this it is easy to show that $QHQ = H$, and so the matrix EEB inequality gives

$$T \log(\Delta) + H \succeq 0. \tag{65}$$

Let $J$ be *the modular involution*, which is the complex-antilnear operator defined as

$$J|a\rangle := |\rho^{1/2} a^* \rho^{-1/2}\rangle. \tag{66}$$

For any $a \in \mathcal{A}$ we have

$$\langle Ja|H|Ja\rangle = \mathrm{tr}(\rho \, \rho^{-1/2} a \rho^{1/2} [h, \rho^{1/2} a^* \rho^{-1/2}]) \tag{67}$$

$$= \mathrm{tr}(a \rho^{1/2} h \rho^{1/2} a^*) - \mathrm{tr}(\rho^{1/2} a \rho a^* \rho^{-1/2} h) \tag{68}$$

$$= - \mathrm{tr}(\rho a^* [h, a]) \tag{69}$$

$$= -\langle a|H|a\rangle, \tag{70}$$

where in the third line we used the fact that $[\rho, h] = 0$. Thus $J^\dagger H J = -H$. The same calculation with $\log(\rho)$ replacing $h$ shows that $J^\dagger \log(\Delta) J = -\log(\Delta)$. It follows that $-(T \log \Delta + H) = J^\dagger (T \log(\Delta) + H) J \succeq 0$ and thus $T \log(\Delta) + H = 0$. From this we see that $\log(\rho) - h/T$ commutes with every operator in $\mathcal{A}$, which means it is a multiple of the identity, and so $\rho = e^{-h/T} / \mathrm{tr}(e^{-h/T})$.

# C   Corrections to ideal algorithm

In this appendix we describe several modifications that were made to the idealized algorithm (8) for the numerical work in section 4.

1. For any matrix $M$, a semidefnite constraint $M \succeq 0$ can be broken down to the constraints $M_- = 0$ and $M_+ \succeq 0$, where $M_\pm := (M \pm M^\dagger)/2$. Doing so with the semidefinite constraint (9) yields

$$\boldsymbol{H}_- = 0 \tag{71}$$

$$\log(\boldsymbol{\Delta}) + \boldsymbol{H}_+ \succeq 0 \tag{72}$$

   Instead of imposing these constraints simultaneously, we impose the linear constraint $\boldsymbol{H}_- = 0$ first. This greatly reduces the number of degrees of freedom in the semidefinite program, leading to a more computationally efficient algorithm.

2. In the presence of noise in the expectation values of $\rho$, it is not appropriate to impose the linear constraint $\boldsymbol{H}_- = 0$ exactly. Instead, the following matrix was computed

$$W_{\alpha\beta} := \mathrm{tr}((\boldsymbol{H}_\alpha - \boldsymbol{H}_\alpha^\dagger)(\boldsymbol{H}_\beta - \boldsymbol{H}_\beta^\dagger)^\dagger), \tag{73}$$

   where $(\boldsymbol{H}_\alpha)_{ij} := \mathrm{tr}(\rho a_i^* [h_\alpha, a_j])$ for $\alpha = 1, \dots s$, and the search space of Hamiltonians was restricted to the span of the eigenvectors of $W$ with eigenvalue below a threshold $\epsilon_W > 0$. This is equivalent to the matrix $\mathcal{K}$ used in [22]. For sufficiently small values of $\sigma_{noise}$, the spectrum of $W$ was found to have several near-zero eigenvalues and a spectral gap to the rest of the eigenvalues, and $\epsilon_W$ was chosen to lie in this gap. An empirical formula that was found to produce an $\epsilon_W$ lying in the spectral gap of $W$ was

$$\epsilon_W = 400 \max(\sigma_{noise}^2 \sqrt{m}, \; 10^{-11}), \tag{74}$$

   where $m$ denotes the number of terms $a_i^* [h_\alpha, a_j]$ such that $[h_\alpha, a_j] \neq 0$. This formula is not expected to be universal across different values of $n$ and choices of perturbing operators. We note that in practice, while choosing $\epsilon_W$ to be too low caused the output to be inaccurate, choosing $\epsilon_W$ to lie above the gap did not significantly affect the accuracy of the result.

3. Although the temperature can be thought of as the same degree of freedom as the overall scale of the Hamiltonian, we chose to explicitly isolate it by adding a variable $T \geq 0$ to the the semidefinite program and replacing the constraint

$$\log(\boldsymbol{\Delta}) + \boldsymbol{H}_+ - \mu I \succeq 0 \tag{75}$$

with

$$T\log(\boldsymbol{\Delta}) + \boldsymbol{H}_+ - \mu I \succeq 0, \tag{76}$$
$$\mathrm{tr}(\rho h) = -1. \tag{77}$$

Here the extra normalization (77) is necessary to eliminate the degree of freedom associated with simultaneously scaling $T, h$, and $\mu$.

4. Although the MOSEK interior-point solver [13] solves the primal and dual programs simultaneously, it was found that the algorithm ran significantly faster when it was called explicitly with the dual program instead of the primal.