# Peer review of "Efficient Hamiltonian learning from Gibbs states"

_SciPost Physics_

## Round 1 · Referee Report · Anonymous (Referee 1) · 2024-12-24

Strengths

  1. The author provides a conceptually new approach for Hamiltonian learning, based on the EEB inequality.

  2. Numerical experiment suggests that this new approach has strong practical potential. (The known provable result in the literature works in theory, but in practice it is not expected to work well.)

  3. The author includes a well-documented github repository.

Weaknesses

How well the method will work in the presence of noise is not discussed.

Report

This is a short and simple paper that contains a fresh new idea on Hamiltonian learning. Its main weakness is that, unlike the other known existing method, the author does not provide a rigorous guarantee in the presence of noise. However, the author makes a partial progress in a followup work (arXiv:2410.23284) with others. Since proposed method has shown promises (both in terms of this recent theoretical progress as well as the shown numerics), there is a good potential for this method to lead to more studies in the future. On that ground, I recommend publication.

Requested changes

  1. When I ran the tester.py using setup.yml, MOSEK complained that the license could not be located. Upon going to the website, it seems that one needs to obtain an academic license. Is this the case? I don't doubt the correctness, but it would be helpful if the author can clarify.

  2. In Proposition 3, I believe \mathcal{S} should be \mathscr{S}.

  3. p.12: antiselfadjoint -> anti-self-adjoint

  4. p.17: the the -> the

Recommendation

Publish (easily meets expectations and criteria for this Journal; among top 50%)

---

## Round 1 · Referee Report · Anonymous (Referee 2) · 2025-1-4

Strengths

  1. The paper establishes a new lower bound on the derivative of entropy resulting from local perturbations of a quantum state. This theoretical result is noteworthy and may be of independent interest beyond the scope of the algorithm itself.
  2. The Hamiltonian learning algorithm proposed in the paper is efficient, with numerical validation demonstrated on a large-scale system involving a 100-qubit Hamiltonian.

Weaknesses

  1. The paper lacks a rigorous proof of the algorithm’s correctness in scenarios where local observables are estimated inaccurately. Previous works (Refs. [2, 18]) devote significant efforts to demonstrating the robustness of their methods under such conditions. However, this manuscript does not mention the issue in the introduction or its comparison with prior work, only briefly mentioning it in the outlook section.

  2. Even when all local observables are estimated accurately, the algorithm identifies a local Hamiltonian that satisfies the matrix EEB inequality. If understood correctly, the matrix EEB inequality is merely a semidefinite relaxation of the EEB condition. As a result, the output Hamiltonian may deviate from the true Hamiltonian. While Appendix A compares this relaxation with the sum-of-squares relaxation in Ref. [18], the authors of Ref. [18] provide rigorous proof that their output Hamiltonian (after relaxation) closely approximates the true Hamiltonian—an assurance not offered in the current work.

  3. The numerical demonstrations provided in the paper are limited in scope. The experiments focus solely on a highly symmetric Heisenberg model, which may not fully test the algorithm's capabilities. It would strengthen the manuscript to include tests on more general Hamiltonians, such as Heisenberg models with random couplings.

Report

Despite the noted weaknesses, I recommend this manuscript for acceptance. Hamiltonian learning is an essential problem in quantum science. The proposed Hamiltonian learning algorithm represents a significant step forward in addressing the scalability challenges of current methods. However, the authors should explicitly acknowledge and emphasize the limitations of their theoretical guarantees, particularly regarding robustness to noise and the relaxation of the EEB condition.

Requested changes

  1. In page 2, "where M anti-Hermitian" should be "where M is anti-Hermitian".
  2. In page 5, "at most k+2k" should be "at most k'+2k".
  3. As mentioned in the report, the authors should discuss the limitations of their theoretical guarantees in the main text.

Recommendation

Ask for minor revision

---

## Round 1 · Referee Report · Anonymous (Referee 3) · 2025-1-15

Report

The author presents a novel algorithm for Hamiltonian learning that appears both original and potentially significant. It derives new bounds on entropy changes under Lindblad evolution and, on this basis, proposes a conceptually new algorithm (incorporating a semidefinite program) for Hamiltonian learning. Numerical simulations illustrate the approach. However, in its current form, certain aspects remain unclear to this referee, making a final assessment difficult. This concerns in particular the following points:

1. Locality Assumptions
The introduction and numerical experiments indicate that the algorithm targets local Hamiltonians and learns from local expectation values. However, it is not clear how locality is used in the theoretical statements. Does the algorithm require that the target Hamiltonian be local? Does its performance degrade with increasing non-locality? Are the Lindbladian operators assumed to be (geometrically) local? What about the classical post-processing cost?

2. Minimization Procedure
The details of the minimization step in the algorithm are not entirely transparent. Is the algorithm varying each local term h fully, or only a single parameter in each term (as the numerical section seems to imply)? It would help to include a general expression of the candidate Hamiltonians that are being optimized.

3. Measurement Noise and Sample Complexity
Beyond a few numerical experiments, the paper does not address how measurement noise or finite sample complexity influences performance. A complete analytical treatment might be outside the scope here, but additional numerical studies (e.g., exploring scaling with system size or the number of parameters) would be beneficial.

4. Conclusion/Outlook with New Content
The conclusion appears to introduce new material (or at least new names)—such as the “recoverability statement” and proposition 4—that is not discussed in the main text. This makes the paper hard to follow without consulting the appendix. In addition, the conclusion should not contain new concepts or results. Instead, these points (especially recoverability and its limitations) should be included and properly explained in the main text so that it remains self-contained.

Overall, the work is promising but would benefit from clarifications about locality, a more explicit description of the optimization step, a discussion (or at least numerical evidence) of the algorithm performance with respect to measurement noise, system size, number of parameters, ... , and a more streamlined conclusion. With these points addressed, the paper could be a very valuable contribution to the field of Hamiltonian learning.

Requested changes

Additional Minor Comments

  1. Appendix Proofs and Main Text Alignment The proofs provided in the appendix could be better linked to the statements in the main text. Some results are labeled “Theorem” in the main text but appear as “Proposition” in the appendix. Please specify which theorem in the main text corresponds to which proposition in the appendix, or adjust the naming scheme to ensure consistency.

  2. Clarification on Perturbation Operators In the numerical results and conclusion, “perturbation operators” are mentioned. It is unclear whether these operators b coincide with the operators a introduced earlier.

  3. Noise Parameter sigma The manuscript mentions a noise parameter sigma. Does this parameter represent the variance or the standard deviation of the Gaussian noise? Also, please specify whether the Gaussian noise is applied independently to each expectation value.

  4. Terminology for “Plateaux” The term “plateaux” appears in the text. Unless French usage is intended, the standard English plural would be “plateaus.”

  5. Conclusion Referencing “Proposition 4” The conclusion refers to “Proposition 4,” but there seems to be no Proposition 4 in the main text. In addition, the term “reconfigurability statement” is introduced without prior definition. Please clarify these references and ensure that new terminology is properly defined or explained.

  6. In the second part of the first paragraph in the introduction, it is not clear whether these are new, original results or results from previous works (in which case references should be added). Also in the second paragraph, it is not quite clear what is meant with the "entropy" (this also concerns the abstract).

Recommendation

Ask for major revision

---

## Editorial Decision

awaiting_resubmission